# Compositionally Disordered Ceramic (Gd,Y,Tb,Ce)$_3$Al$_2$Ga$_3$O$_{12}$ Phosphor for an Effective Conversion of Isotopes' Ionizing Radiation to Light

**Mikhail V. Korzhik** [1,2,*], **Petr V. Karpyuk** [1], **Aliaksei G. Bondarau** [2], **Daria E. Lelecova** [1], **Vitaly A. Mechinsky** [1,2], **Vladimir Pustovarov** [3], **Vasilii Retivov** [1], **Valentina G. Smyslova** [1], **Dmitry Tavrunov** [3] and **Denis N. Yanushevich** [4]

1  National Research Center "Kurchatov Institute", 123098 Moscow, Russia; silancedie@mail.ru (P.V.K.); daria_kyznecova@inbox.ru (D.E.L.); vitaly.mechinsky@cern.ch (V.A.M.); vasilii_retivov@mail.ru (V.R.); smyslovavg@gmail.com (V.G.S.)
2  Institute for Nuclear Problems, Belarus State University, 11 Bobruiskaya, 220006 Minsk, Belarus; alesonep@gmail.com
3  Experimental Physics Department, Ural Federal University, 620075 Yekaterinburg, Russia; vpustovarov@bk.ru (V.P.); mr.tavrunov@mail.ru (D.T.)
4  Physical Department, Belarus State University, 4 Independence Av., 220006 Minsk, Belarus; denisn.yanychevic@gmail.com
*  Correspondence: mikhail.korjik@cern.ch

**Abstract:** Compositionally disordered crystalline material (Gd,Y,Tb,Ce)$_3$Al$_2$Ga$_3$O$_{12}$ was demonstrated to be a highly effective converter of corpuscular ionizing radiation into light. The material was found to be radiation-tolerant to an intense 10 MeV electron beam and had a low temperature dependence on light yield. These findings open an opportunity to utilize the developed material to create long-living, high-flux sources of optical photons under the irradiation of isotope sources. Besides the purposes of the measurement of ionizing radiation by the scintillation method in a harsh irradiation environment, this puts forward the exploiting of the developed material for indirect isotope voltaic batteries and the consideration of a photon engine for travel beyond the solar system, where solar wind force becomes negligible.

**Keywords:** compositional disorder; ceramics; garnet structure; terbium; cerium; scintillation light yield; radiation damage; indirect isotope radiation convertor

## 1. Introduction

Scintillation materials are widely used in ionizing radiation detectors [1]. Among the many luminescent materials utilized for this purpose, inorganic crystalline scintillation materials stand out because they provide high stopping power. At the same time, the increase in the nomenclature of such materials is significantly constrained by the limited number of methods for their preparation in crystalline form. Practically any inorganic scintillator can be produced in powder form by coprecipitation [2], solid-phase synthesis [3], and pyrolysis [4,5]. However, the set of methods that ensures the rapid transformation of raw materials into a single crystalline mass is limited [6]. Moreover, the growth of crystals with an increase in the number of cations in a compound of more than four inevitably encounters problems with different evaporation rates of the components and, consequently, an increased rate of defect formation in the crystalline mass [7]. An increased concentration of defects leads to a deterioration in the user properties of materials under the influence of ionizing radiation, especially radiation resistance and phosphorescence, which creates additional optical noise.

Production of scintillation materials in the form of ceramics is becoming more and more widely used both in research and in the production of scintillation elements [8]. Although the number of compounds available to obtain high-quality transparent ceramics

is limited by the crystalline materials with spatial cubic symmetry, its production is not burdened by the use of expensive tooling based on precious metals. Moreover, ceramic elements can be produced in complex shapes and sizes, which is not achievable with crystal pulling techniques.

The methods of ceramic production make it possible to accurately control the composition of a compound even with a large number of cations in the matrix. These methods are well developed for systems with a garnet structure [9]. Moreover, complicated garnet compounds can be created by mixing and further sintering the less complex compounds of the same structural type as well. An increase in the number of cations in the garnet crystal lattice inevitably leads to compositional disordering of the crystal system [10]. The varying lattice composition and the concentration of activator centers make it possible to tune the properties of the material for a specific application [11–19].

In this work, we set the goal of creating a scintillator with a high yield and radiation resistance not only for detection purposes but also for obtaining maximum optical photon fluxes when converting the energy of absorbed charged particles in the visible spectral range. The activator providing high excitation efficiency in the compositionally disordered garnet compound $(Gd,Y)_3Al_2Ga_3O_{12}$ is the $Tb^{3+}$ ion [20], which isovalently substitutes for the Gd and Y ions in the matrix. $Tb^{3+}$ ions possess slowly decaying luminescence but with a high luminosity [21–23] and cathodoluminescence yield [24]. However, the emission spectrum of $Tb^{3+}$ ions is formed by a set of narrow luminescence bands due to the $^5D_{3,4}\rightarrow^7F_J$ transitions. Coactivation with Ce ions is used to increase the spectral density of luminescence in the visible range. Materials doped with Tb and Ce ions are also the subject of lively discussion for the purposes of LED lighting [25–31] and photoconversion [32]. Nevertheless, the use of codoping by the Ce and Tb ions requires a carefully chosen excitation length for the simultaneous effective excitation of the luminescence of both activators. When excited by ionizing radiation, the Ce and Tb ions quite effectively interact with nonequilibrium carriers [33], thereby providing the addition of luminescent fluxes. Moreover, the Ce and Tb ions also interact with each other; the luminescence of the $Ce^{3+}$ ions is sensitized by the $Tb^{3+}$ ions and vice versa [34–39].

For the new promising applications, the radiation resistance of the material, combined with the thermal stability of the scintillation yield, are critical parameters. Crystalline materials activated with both cerium [40] and terbium [41–43] are tolerant to exposure to the electromagnetic component of ionizing radiation and high-energy protons. Codoping with $Ce^{3+}$ and $Tb^{3+}$ ions increases the radiation resistance of materials [44]. Among materials with compositional disorder, the highest tolerance to irradiation with charged particles was measured for crystalline compounds of the family of gadolinium–aluminum–gallium garnets $Gd_3Al_2Ga_3O_{12}$ (GAGG) and $(Gd,Y)_3Al_2Ga_3O_{12}$ (GYAGG) [45,46]. It is caused by the obstruction to the formation of the metastable trapping centers that captured nonequilibrium electrons between the bottom of the conduction band and sub-bands formed by the f-states ($^6P$, $^6I$, $^6D$) of $Gd^{3+}$ ions in the band gap. In crystalline compounds, the codoping by $Ce^{3+}$ and $Tb^{3+}$ ions creates a sufficiently dense filling of the bandgap by 4f- and 5d-states, which ensures fast relaxation of nonequilibrium carriers formed by ionizing radiation.

Fine-tuning the efficiency of the transfer of electronic excitations to the $Ce^{3+}$ and $Tb^{3+}$ activator ions is carried out by partial substitution of gadolinium ions by yttrium ions in the matrix host [47]. Managing the high scintillation yield, especially in GYAGG:Tb ceramics, made it possible to expand a number of possible applications of the material, in particular for indirect converters of charged particle radiation from isotope sources into electricity [48].

When using isotopes emitting α-particles, the efficiency of conversion to light in GYAGG:Tb can reach 10% per scintillator plate. For simplicity, if at least half of the photons move in the direction opposite to the plane with the isotope deposited on the scintillator, we can estimate that the pressure created by the system (scintillator-$^{241}$Am) is ~1 pN/cm$^2$;. Having a converter plate with a thickness of no more than 100 microns, which is more than enough to absorb α-particles, and a square of 25 m$^2$, we obtain a resulting force of 250 nN. That is comparable to the force

due to the anisotropic thermal radiation of the Pioneer apparatus, which provided an additional acceleration of $8.74 \pm 1.33 \times 10{-10}$ m/s$^2$; [49,50]. Interestingly, it is not necessary to associate the photon engine only with the annihilation of matter; isotope photon engines are also of interest for movement beyond the solar system.

We focused in this study on optimizing the concentration ratio of Ce and Tb ions in the $(Gd,Y,Tb,Ce)_3Al_2Ga_3O_{12}$ phosphor converter material. This parameter is key to achieving the highest tolerance to ionizing radiation, a high light yield, and a moderate temperature dependence. Besides the highly promising utilizations, the listed combination of parameters opens the door for other applications of the material, including X-ray radiography devices and computed tomography (CT) scanners. The light yield was estimated using several methods, exploiting different excitations ranging from charged particles to hard photons, including X-rays. The conclusions drawn are supported by the results of measuring the properties of luminescence for various types of excitations.

## 2. Materials and Methods

A series of $(Gd,Y,Tb,Ce)_3Al_2Ga_3O_{12}$ samples was produced. For comparison, a sample solely doped with Tb $(Gd,Y,Tb)_3Al_2Ga_3O_{12}$ was manufactured as well. Compositions of the samples are in Table 1. Indexes X in the compound of Gd, Y, Tb, Ce ions can be obtained by multiplying the numbers in the composition by a factor of three.

**Table 1.** Composition of the studied samples.

| No. | Composition |
|:---:|:---:|
| 1 | $(Gd_{0.373}Y_{0.560}Tb_{0.067}Ce_{0.0003})_3Al_2Ga_3O_{12}$ |
| 2 | $(Gd_{0.372}Y_{0.558}Tb_{0.067}Ce_{0.003})_3Al_2Ga_3O_{12}$ |
| 3 | $(Gd_{0.367}Y_{0.550}Tb_{0.067}Ce_{0.016})_3Al_2Ga_3O_{12}$ |
| 4 | $(Gd_{0.387}Y_{0.580}Tb_{0.03}Ce_{0.0003})_3Al_2Ga_3O_{12}$ |
| 5 | $(Gd_{0.385}Y_{0.578}Tb_{0.03}Ce_{0.003})_3Al_2Ga_3O_{12}$ |
| 6 | $(Gd_{0.380}Y_{0.570}Tb_{0.03}Ce_{0.016})_3Al_2Ga_3O_{12}$ |
| 7 | $(Gd_{0.393}Y_{0.590}Tb_{0.016}Ce_{0.0003})_3Al_2Ga_3O_{12}$ |
| 8 | $(Gd_{0.393}Y_{0.590}Tb_{0.016}Ce_{0.003})_3Al_2Ga_3O_{12}$ |
| 9 | $(Gd_{0.393}Y_{0.589}Tb_{0.016}Ce_{0.016})_3Al_2Ga_3O_{12}$ |
| 10 | $(Gd_{0.387}Y_{0.580}Tb_{0.03})_3Al_2Ga_3O_{12}$ |

Figure 1a,b depicts the SEM image of the calcined at 850 °C for 2 h and the representative precursor obtained by the reverse coprecipitation method and the resulting ceramics, namely sample 3. The method of reverse coprecipitation consisted of adding a mixed nitrate solution of the components (Gd, Y, Tb, Ce, Al, Ga) to an excess of ammonium bicarbonate solution ($NH_4HCO_3$) under constant stirring. The resultant precipitate was calcined at 850 °C and compacted by the uniaxial pressing method.

The ceramics samples were produced by sintering in air of the densified pellets in the Nabertherm LHT 02/17 LB furnace (Germany) at 1600 °C for 2 h. All samples had a diameter of 12 mm and were translucent. They were ground and polished to a thickness of 1 mm. After measurements with X-ray and electron beam excitations, samples were cut into two parts each. One part was used for X-ray diffraction spectra measurements. Second halves at different UV excitations are depicted in Figure 1 (bottom panels).

X-ray diffraction spectra measured with a 2D Phaser, Bruker (Germany) (Cu k$\alpha$ doublet 1.5406 and 1.5444 Å) confirmed that codoping of GYAGG crystal does not change the structure of the compound; it remains cubic, as seen from Figure 2.

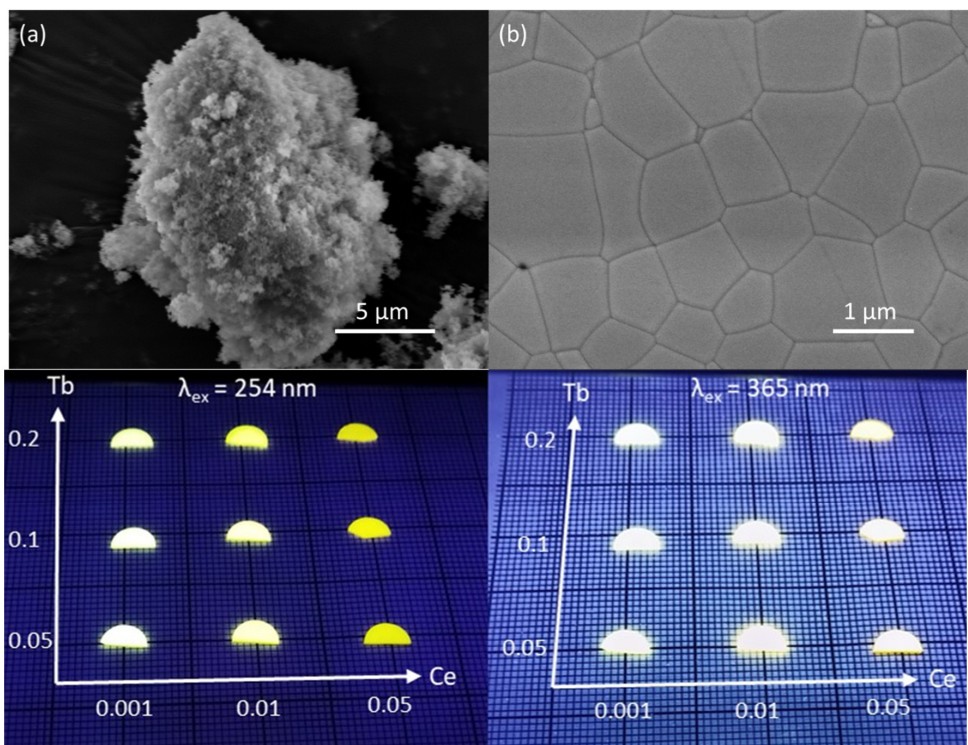

**Figure 1.** SEM images of representative precursor of $(Gd,Y,Tb,Ce)_3Al_2Ga_3O_{12}$ (**a**) and ceramics $(Gd_{0.372}Y_{0.558}Tb_{0.067}Ce_{0.016})_3Al_2Ga_3O_{12}$ (**b**) and samples under UV excitation of different wavelengths (bottom panels).

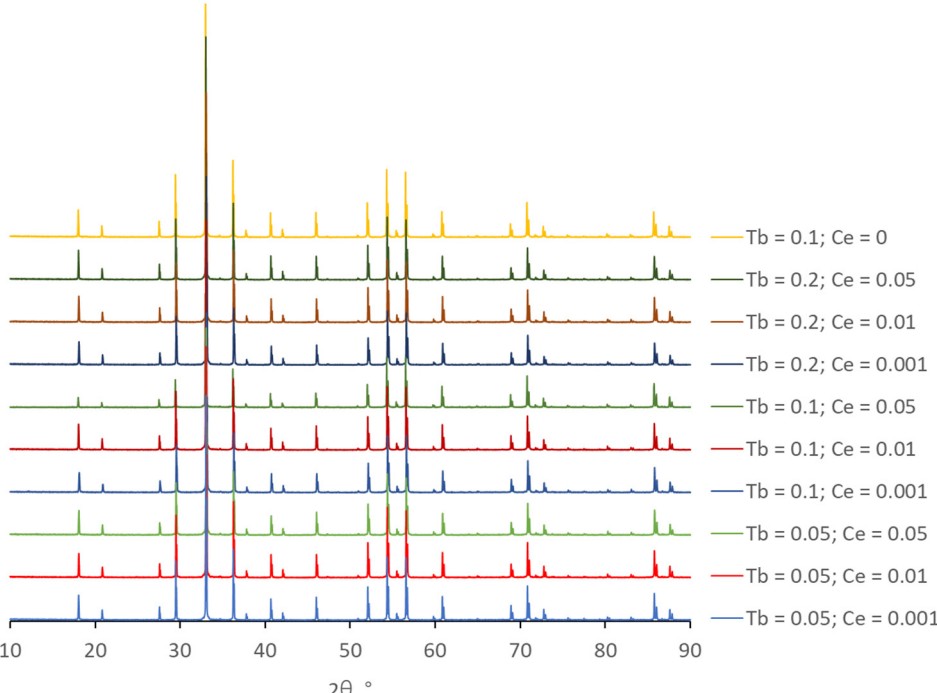

**Figure 2.** X-ray diffraction pattern of GYAGG:Tb,Ce ceramics samples with different terbium and cerium concentrations.

The scintillation yield was evaluated by the following methods. The total scintillation yield of $Ce^{3+}$ and $Tb^{3+}$ ions was measured using Philips XP 2020 photomultiplier (Philips, The Netherlands) in the current mode, as described in [51]. A reference sample

with ground surfaces to mimic translucent sample, a YAG:Ce single crystal possessing a light yield of 4100 ph/MeV under α-particles was used.

In addition, the temperature dependence of the total scintillation yield and the spectral distribution before and after irradiation with a 10 MeV electron beam were measured with X-ray excitation by the URS-55A apparatus (Ekaterinburg, Russia) with the BSV-2 X-ray tube (30 kV, 10 mA, Cu anode). The radioluminescence (XRL) was detected using a LOMO MDR-23 monochromator (LOMO, Russia) with a spectral resolution of 2 nm and a photon-counting FEU-106 photomultiplier tube. The same bench was used to evaluate thermostimulated luminescence (TSL) above room temperature.

The total scintillation yield for γ-quanta was estimated from the obtained data with α-particles by using the α/γ ratio. For a YAG:Ce single crystal, the α/γ ratio was selected to be 0.2 [52,53]. For GYAGG:Ce, this coefficient was taken to be equal to 0.24. For terbium-activated samples, the values of the α/γ ratio determined in [49] were used.

Irradiation with fast electrons (E = 10 MeV) was performed at the UELR-10-10C2 linear electron accelerator (Ekaterinburg, Russia). To avoid samples heating under the irradiation beam, an existing conveyor system with partial irradiation doses of 11 kGy per cycle was used [54]. Measurement of the total absorbed dose during the exposure was carried out using a certified film dosimeter SO PD (E)-1/10 "VNIIFTRI" by measuring the induced optical density at a wavelength of 550 nm. The total absorbed dose was 105 kGy.

## 3. Results and Discussion

The XRL spectra of the samples are shown in Figure 3. The spectrum is a superposition of $Ce^{3+}$ and $Tb^{3+}$ ion bands: a wide $Ce^{3+}$ interconfiguration luminescence overlaps narrow intraconfiguration luminescence bands of $Tb^{3+}$ ions.

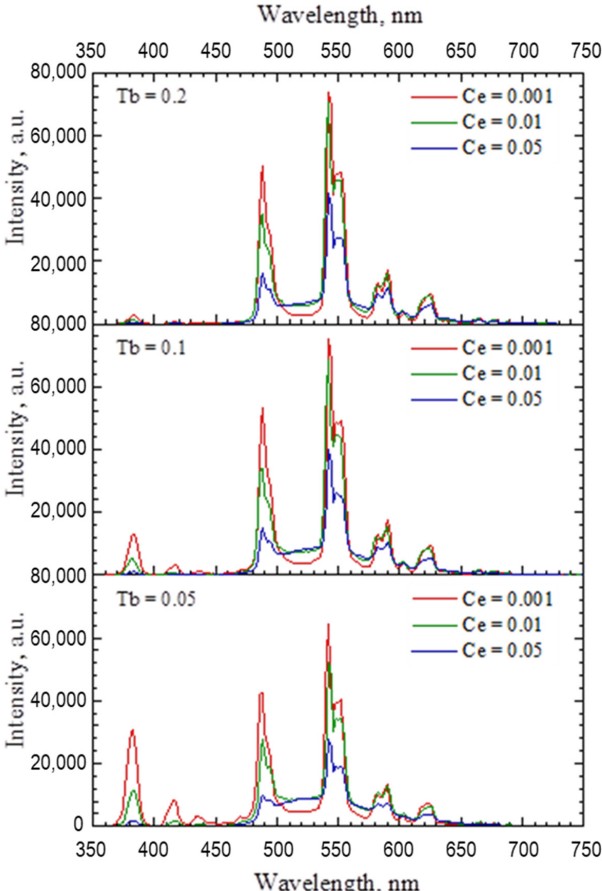

**Figure 3.** XRL spectra of the samples measured at room temperature. Indexes X of Ce and Tb in the compound indicated.

The contribution of the cerium band increases with an increase in its concentration. At the lowest Ce concentration, its contribution is small, which indicates that the $Tb^{3+}$ ions catch most of the matrix excitations. However, at Ce concentration x = 0.01, its proportion increases and further stabilizes. Apparently, the optimal Ce concentration is close to x = 0.01. With an increase in the Tb concentration, the intensity of the bands in the blue region of the spectrum is redistributed in favor of the bands in the green–yellow region, which is due to the concentration quenching of the $^5D_3 \rightarrow {}^7D_J$ transition [55].

A comparison of their integrated intensities in the spectral range of 360–750 nm is shown in Figure 4. The points with the same concentration of Tb are connected by straight lines on the plots of the dependence of the integral intensities for better perception. As seen, it is not advisable to increase the concentration of terbium more than x = 0.2, and the optimum concentration in terms of light output is close to x = 0.1.

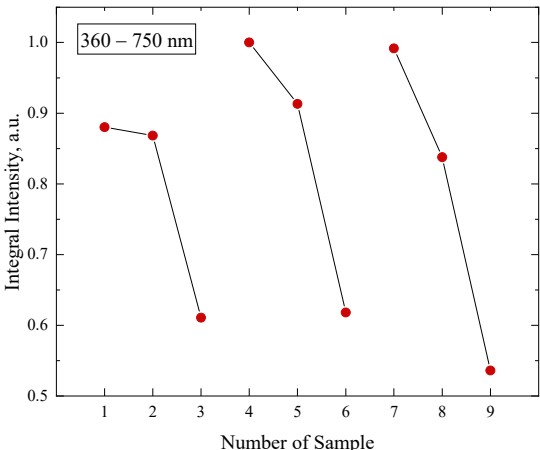

**Figure 4.** Samples normalized X-ray excited luminescence integral intensity measured in the spectral range of 360–750 nm.

The temperature dependence of the XRL intensity for samples with an optimal Ce content is shown in Figure 5.

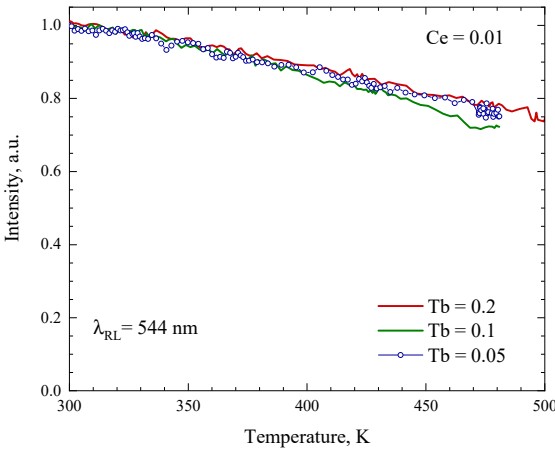

**Figure 5.** Normalized temperature dependence of the XRL intensity at the registration at 544 nm for samples with an optimal Ce content. Concentrations of Ce and Tb indicated.

It is known that crystals of the GAGG family, upon activation with $Ce^{3+}$ ions, have a high temperature coefficient of scintillation yield, while upon activation with $Tb^{3+}$, it is very small in the temperature range of up to 500 K [23]. The superposition of the XRL temperature dependences of Ce and Tb ions forms the observed results. In addition,

the temperature increase accelerates the electronic excitation transfer from Tb to Ce ions, which is followed by nonradiative relaxation at high temperatures in the latter. At the same time, we note that the decrease in intensity upon heating to 500 K does not exceed 25%.

For the sample solely doped with Tb and samples with the optimal Ce content and in the Tb concentration range, the XRL spectra were measured before and after irradiation by a 10 MeV electron beam with a total absorbed dose of 105 kGy. A comparison of the spectra is shown in Figure 6. No change in the XRL spectral distribution was found. A change in the XRL integrated intensity for a Tb-doped sample was estimated to be 0.3%, whereas it was defined as small as 0.1% for codoped samples. The latter estimation is within the error of the measurements and might be even less. This confirms the high stability of the parameters of the developed converter materials for operation with isotope sources.

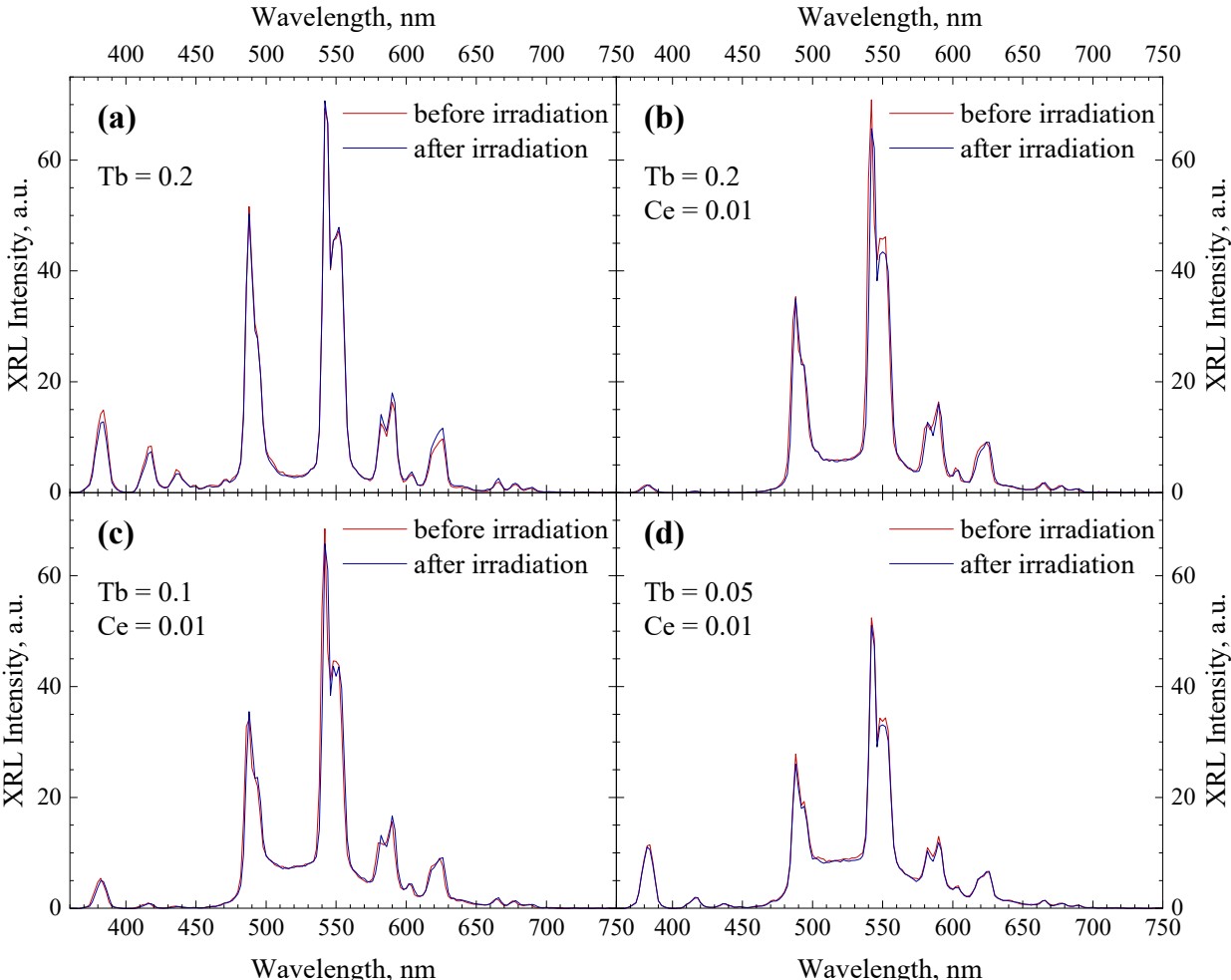

**Figure 6.** A comparison of the XRL spectra of samples without Ce (**a**) and representative samples with an optimal Ce content and in the Tb concentration range (**b–d**) before and after irradiation with a 10 MeV electron beam.

The radiation resistance of the material was measured under irradiation with a 10 MeV electron beam. Electrons of such energy damage the lattice due to the displacement of atoms into interstices in the crystal lattice [1]. From the point of view of using the phosphor as a converter, the concentration of the point structure defects is crucially important. They lead to the creation of deep trapping centers, i.e., they ensure the redistribution of the stored energy of electronic excitations to the phosphorescence region. Temperature-stimulated luminescence (TSL) spectra were measured in the temperature range in which the temperature dependence of the XRL yield was measured. Since terbium ions make

the main contribution to radioluminescence intensity, the TSL spectra were measured at a wavelength of 544 nm. The results of a comparison of TSL curves measured with a nonirradiated sample with the smallest Ce content and with an optimal Ce content after irradiation with 10 MeV electrons to an absorbed dose of 105 kGy are shown in Figure 7. Before measuring TSL, the samples were irradiated with X-rays for 20 min to a dose of 525 Gy. No new quasistationary trapping centers were found that could potentially provide an increase in the phosphorescence yield.

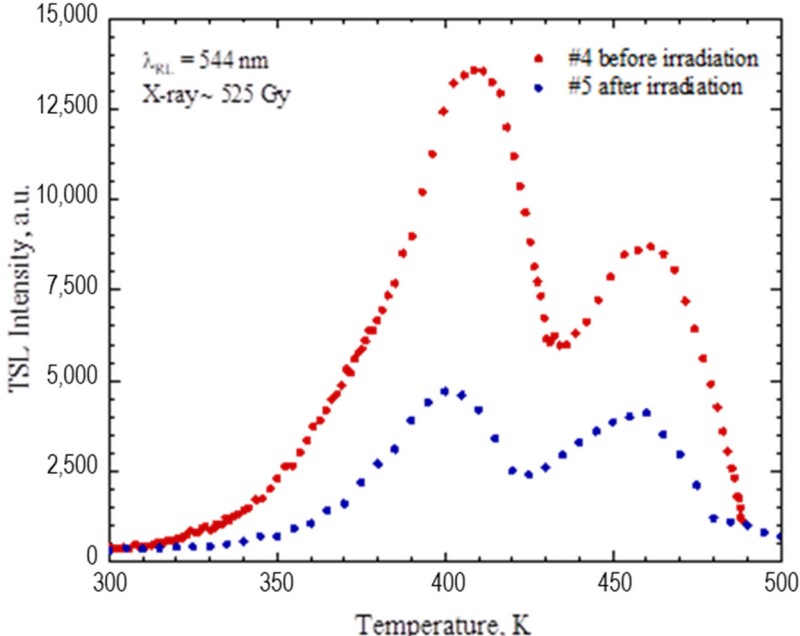

**Figure 7.** Thermally stimulated luminescence of $(Gd,Y,Tb,Ce)_3Al_2Ga_3O_{12}$ representative sample #4 before irradiation and #5 after irradiation with a 10 MeV electron beam. Linear heating rate: 0.33 K/s.

The results of measuring the light yield upon excitation with $\alpha$-particles are depicted in Figure 8a. Straight lines on the plots connect the points with the different concentrations of Tb for better perception.

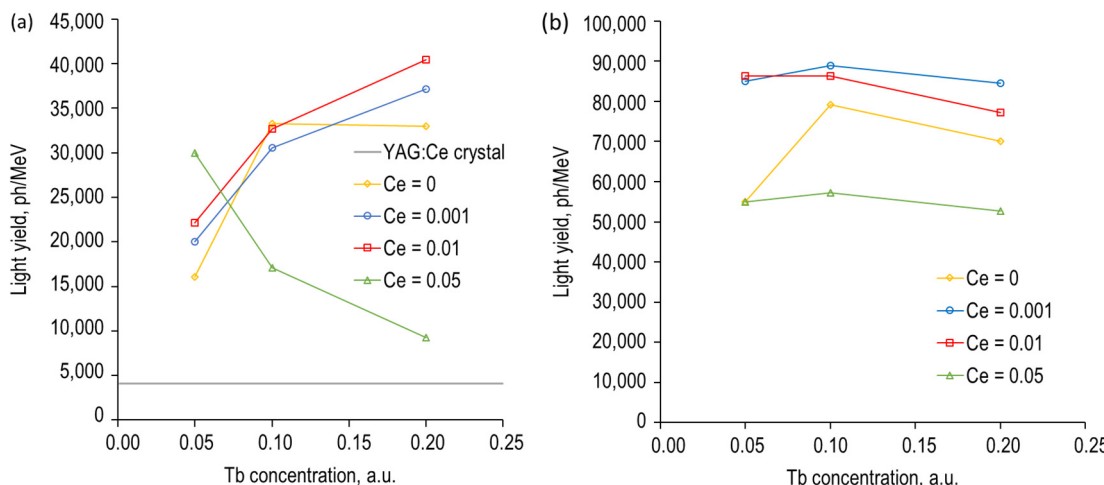

**Figure 8.** The total light yield measured at room temperature at excitation with $\alpha$-particles of $^{241}$Am source (**a**) and calculated for $\gamma$-quanta excitation by using the $\alpha/\gamma$ ratio (**b**).

As seen, coactivation with Ce ions smooths out the dependence of the yield of scintillation at high concentrations with terbium ions. This effect is due to the sensitization

of the luminescence of $Tb^{3+}$ ions by $Ce^{3+}$ ions. A difference in the behavior of the light yield at the highest Ce concentration, apparently, is caused by combining the concentration quenching of $Ce^{3+}$ luminescence with high transfer efficiency from $Tb^{3+}$ to $Ce^{3+}$ ions due to a shortening of the average distance between Ce and Tb ions in the lattice. For typical isotopes that emit $\alpha$-particles and are of practical importance, the energy of an $\alpha$-particle emitted from a thin layer is ~5.5 MeV. Thus, at least 220,000 photons will be emitted when one $\alpha$-particle is absorbed.

To estimate the yield of scintillations at the registration of electromagnetic particles, the previously indicated numbers of $\alpha/\gamma$ were used. The evaluation results are shown in Figure 8b. The results of the estimation correlate with the data presented in Figure 4; however, the sample with the highest concentration of Ce demonstrated a difference at the level of 40% compared with other samples. Apparently, this difference is caused by the sensitivity of the method used to the $\alpha/\gamma$ ratio. Nevertheless, the codoped samples show better light yield than the solely Tb-doped sample. Thus, besides $\alpha$-particles, $(Gd,Y,Tb,Ce)_3Al_2Ga_3O_{12}$ is an efficient converter for excitation by $\beta$-emitting isotopes as well.

The longevity of the conversion material developed was estimated. Table 2 lists some parameters of isotope sources that can be deposited on converter material plates. Table 2 demonstrates that activities and energy deposits from the isotopes suitable for $\alpha$- and $\beta$-irradiation conversion are comparable with the irradiation conditions used.

**Table 2.** Some features of commonly used isotope sources.

| Isotope | $^{238}Pu$ | $^{241}Am$ | $^{147}Pm$ |
|---|---|---|---|
| Emitted particles | $\alpha$-particle emitter | $\alpha$-particle emitter | $\beta$-particle emitter |
| Half-life, year/s | $87.7/2.77 \times 10^9$ | $432.2/1.36 \times 10^{10}$ | $2.62/8.26 \times 10^7$ |
| Radiation energy, eV (branching, %) [56] | $5.499 \times 10^6$ (71) $5.456 \times 10^6$ (29) | $5.486 \times 10^6$ (85) $5.443 \times 10^6$ (13) $5.388 \times 10^6$ (2) | $E_{max}$ = 224.5 keV (100%) $<E>$ = 61.78 keV |
| Density in metallic form, $g/cm^3$ | 19.8 | 13.8 [57] | 7.26 |
| Activity of 1 $cm^3$, Bq/Ci | $12.53 \times 10^{12}/339$ | $1.76 \times 10^{12}/47.6$ | $2.50 \times 10^{14}/6756.8$ |
| Released energy from 1 $cm^3$, W | 11.1 ($\alpha$-particles only) | 1.38 ($\alpha$-particles only) | 3.25 |
| Released energy from 1 g, W | 0.56 ($\alpha$-particles only) | 0.10 ($\alpha$-particles only) | 0.46 |
| Released energy from a radioisotope foil with an area of 1 $cm^2$ and a thickness of 1 $\mu m$, W | $11.1 \times 10^{-4}$ | $1.38 \times 10^{-4}$ | $3.34 \times 10^{-4}$ |

The profiles of the energy losses in $(Gd,Y,Tb,Ce)_3Al_2Ga_3O_{12}$ material for 5.5 MeV $\alpha$-particles and $\beta$-particles having a spectrum of $^{147}Pm$ [58] are shown in Figure 9.

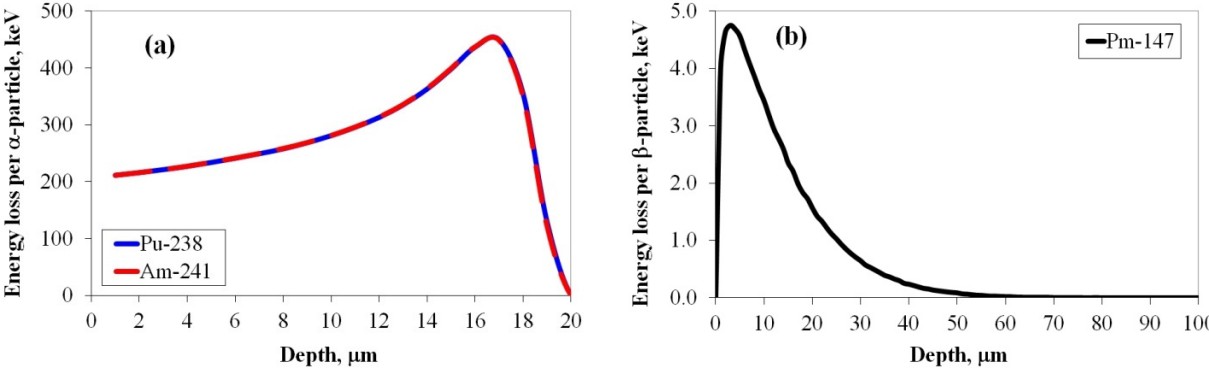

**Figure 9.** Energy loss profile for $\alpha$-particles (**a**) and $\beta$-particles (**b**) of specified isotopes in $(Gd,Y,Tb,Ce)_3Al_2Ga_3O_{12}$ converter material.

When using a converter with an area of 1 $cm^2$ and a density of 6 $g/cm^3$, $\alpha$-particles with an energy of 5.5 MeV are absorbed with an average range of approximately 17 $\mu m$ (90% of the energy deposition), and the mass of the layer subjected to the radiation damage is $10^{-5}$ kg. The 17 $\mu m$ scintillator layer is affected by less than half of the energy released from the 1 $\mu m$ thick radioisotope foil since (a) $\alpha$-particles exit the foil in both directions,

(b) approximately 10% of the initial energy is absorbed by the foil itself, and (c) 10% of the initial energy in the case of α-particles is carried away by scintillation photons (at a light output of 40,000 ph/MeV and an average photon energy of 2.5 eV). Therefore, radiation damage of the convertor material by α-particles is created by an energy equal to that of $3.5 \times 10^{-4}$ ($^{238}$Pu) and $0.43 \times 10^{-4}$ W ($^{241}$Am).

In the case of $^{147}$Pm, 50% of the electrons will exit in the opposite direction from the convertor, absorption in a 1 μm thick foil is less than 5%, and photons will carry away (at a light output of 85,000 ph/MeV and an average photon energy of 2.5 eV) ~20% of the initial energy. Therefore, the radiation damage of the converter under $^{147}$Pm is created by an energy equal to $1.2 \times 10^{-4}$ W.

Thus, the expected radiation load provided by α-particles on the 17 μm thick converter layer is 35 Gy/s and 4.3 Gy/s for $^{238}$Pu and $^{241}$Am, respectively. Finally, 90% of the β-particles energy release will be reached at a depth of less than 29 μm when $^{147}$Pm is used. The mass of the layer subjected to the radiation damage is $1.7 \times 10^{-5}$ kg. The expected radiation load on the 29 μm scintillator layer is 7.1 Gy/s. The light yield of the convertor was estimated to be a little degraded (~0.1%) at the absorbed dose of 105 kGy. Assuming that the operation time of the converter is limited by a 50% degradation, one can estimate the minimal longevity of the converter material as follows: at the use of $^{238}$Pu, $^{241}$Am, and $^{147}$Pm, it is ~400, 3400, and 2000 h, respectively. Such an assessment is indeed minimal since it does not take into account two important aspects. First, both alpha and beta particles of the isotope sources considered in the study do not produce new defects in the crystal lattice. The number of defects existing in ceramics is limited; therefore, the level of induced absorption due to the limited number of color centers can relatively quickly reach the level of dynamic saturation, which will cause degradation of the light output much less than the estimated level. Moreover, the converter material will heat up during operation, which will undoubtedly lead to thermal annealing of the color centers. Most of the trapping centers will be annealed at temperatures below 450 K, as seen from the high-temperature TSL data.

Considering the assembly of the converter isotope for a photon engine, one must keep in mind that an additional impulse will also be from the pressure of solar wind protons on the sail of the converter material, of course, if it moves in the direction opposite to the sun. The pressure of the sun's wind in the region of the Earth is about 1 nN [59]. The converter provides 1 pN force per 1 cm$^2$, i.e., 1000 times smaller. According to the law $1/R^2$, the sail can operate on its own power already at 32 astronomical units, i.e., beyond the orbit of Neptune.

## 4. Conclusions

A (Gd, Y, Tb, Ce)$_3$Al$_2$Ga$_3$O$_{12}$ ceramic phosphor was, for the first time, developed and optimized for use as a converter of radiation from isotope sources into a beam of optical photons. It was shown that the developed material combines several unique properties, namely, high radiation resistance when absorbing corpuscular radiation, high light yield, and high stability with increasing temperature.

For typical isotopes that emit α-particles with an energy ~5.5 MeV, the photon flash at the level of 220,000 ph from the convertor material will be emitted when one α-particle is absorbed. At the same time, the decrease in the photon number in the flash upon heating to 500 K does not exceed 25%.

Apparently, a combination of Ce and Tb concentrations at the levels x = 0.01 and x = 0.1–0.2, respectively, is optimal to provide the best luminosity of the material.

Samples with optimal Ce and Tb concentrations were subjected to exposure to a 10 MeV electron beam with an absorbed dose of 105 kGy. Practically, no visible changes in the spectral distribution or intensity were detected. A comparison of the spectra showed that the change in the radioluminescence-integrated intensity is less than 0.1%. This allowed, for the first time, an estimation of the minimal longevity of the converter material with the use of $^{238}$Pu, $^{241}$Am, and $^{147}$Pr isotope sources.

**Author Contributions:** Conceptualization, M.V.K. and V.R.; methodology, P.V.K. and V.G.S.; validation, V.P.; formal analysis, D.N.Y.; investigation, A.G.B., D.E.L., V.A.M., D.T. and V.G.S.; data curation, V.A.M.; writing—original draft preparation, M.V.K. and P.V.K.; writing—review and editing, D.E.L. and V.P.; visualization, V.G.S. and V.A.M. All authors have read and agreed to the published version of the manuscript.

**Funding:** This research received no external funding.

**Institutional Review Board Statement:** No institutional review board statement is required.

**Informed Consent Statement:** Not applicable.

**Data Availability Statement:** No new and additional data are available.

**Acknowledgments:** The work was supported by the National Research Center "Kurchatov Institute". Analytical research was conducted using the equipment of the Research Chemical and Analytical Center NRC, Kurchatov Institute, Shared Research Facilities, under the project's financial support by the Russian Federation, represented by The Ministry of Science and Higher Education of the Russian Federation, Agreement No. 075-15-2023-370 dd. 22.02.2023. The research at Ural Federal University was partially supported by the Russian Ministry of Science and Education, Project FEUZ 2023-0013.

**Conflicts of Interest:** The authors declare no conflict of interest.

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
