# Peer review of "Compositionally Disordered Ceramic (Gd,Y,Tb,Ce)3Al2Ga3O12 Phosphor for an Effective Conversion of Isotopes’ Ionizing Radiation to Light"

_ceramics, doi:10.3390/ceramics6030117_

Round 1
Reviewer 1 Report
1. Add references to the statement “Moreover growth of the……….in the crystalline mass.” Line -35.
2. Add references to the statement “The varying lattice composition ………. a specific application.” Line -53.
i.e. 10.3390/photonics8030071
3. XRD is very important data. Add xrd pattern of all the composition used in this work (Table 1).
4. Explain briefly the reverse co-precipitation method for the readers.
5. In experimental section include the precursors salts used for the synthesis of various samples.
6. Figure 2, The emission is from Tb3+ at various Ce3+ concentration. The change is not obvious. mark the emission peak from Ce3+ .
7. Figure 2 captions; include the composition of the samples used for collecting the data.
8. Figure 3 caption is not well written, rewrite the caption explaining what the data is representing .
Author Response
|
The Reviewer comment |
Author response |
|
Add references to the statement “Moreover growth of the……….in the crystalline mass.” Line -35. |
Reference added. |
|
Add references to the statement “The varying lattice composition ………. a specific application.” Line -53 i.e. 10.3390/photonics8030071 |
New references have been added. |
|
XRD is very important data. Add xrd pattern of all the composition used in this work (Table 1). |
We agree with the Reviewer that this information is important, XDR patterns have been added. |
|
Explain briefly the reverse co-precipitation method for the readers. |
New appropriate sentences have been included in the text. |
|
In experimental section include the precursors salts used for the synthesis of various samples. |
They are included in the new sentences, briefly describing the inverse coprecipitation. |
|
Figure 2, The emission is from Tb3+ at various Ce3+ concentration. The change is not obvious. mark the emission peak from Ce3+ . |
Ce3+ emission is characterized by a wide luminescence band overlapping Tb3+ narrow bands. In fact, it is observed between these narrow bands, as seen from Fig. 3 in the revised manuscript.
|
|
Figure 2 captions; include the composition of the samples used for collecting the data. |
To avoid overloading the figures, we prefer to keep different colors and indexes of the Ce and Tb in the compound. As mentioned, Indexes X in the compound of Tb and Ce ions can be obtained by multiplying the numbers in the composition by a factor of three in Table 1.
|
|
Figure 3 caption is not well written, rewrite the caption explaining what the data is representing |
Figure 4 caption is rewritten in the revised manuscript. |

Reviewer 2 Report
The authors submitted a study on the scintillating properties of Ce,Tb-codoped GYAGG translucent ceramic. The article is sufficiently well written and consistent. The results are interesting and conclusions are mostly correct. I recommend this article to be published following minor revision. Here is a list of my comments:
1. The authors should cite their another paper on Ce- and Tb-Doped GYAGG: DOI: 10.3390/app13053323
2. Photographs of translucent ceramics could be included in the manuscript or Supplementary Materials. Ideally photographs of the samples before and after irradiation.
3. Fig. 6 – was the reduction of TSL intensity observed after irradiation? If so, what was the reason for this reduction. If not, why is not the intensity normalized?
4. The application of the studied material as a photon engine has not yet been demonstrated to be feasible and therefore such conclusions should be removed from the manuscript.
Minor English corrections are required.
Author Response
|
The Reviewer comment |
Author response |
|
The authors should cite their another paper on Ce- and Tb-Doped GYAGG: DOI: 10.3390/app13053323 |
Thank you for your interest in our articles. We got a notification from the Editor about an overselfciting. Due to this reason we prefer to cite in in next publications.
|
|
Photographs of translucent ceramics could be included in the manuscript or Supplementary Materials. Ideally photographs of the samples before and after irradiation.
|
The photographs of the samples under different UV excitations are added to the revised manuscript.
|
|
Fig. 6 – was the reduction of TSL intensity observed after irradiation? If so, what was the reason for this reduction. If not, why is not the intensity normalized?
|
Yes, the TSL intensity was reduced after irradiation. Electrons of selected energy maydamage the lattice due to the displacement of atoms . Perhaps some of these knocked-out atoms diffuse and fill lattice defects created during the samples preparation. This observation should be studied carefully and will be the subject of a separate publication.
|
|
The application of the studied material as a photon engine has not yet been demonstrated to be feasible and therefore such conclusions should be removed from the manuscript. |
This is removed from the conclusion chapter in the discussion part.
|
|
Minor English corrections are required |
The manuscript has been checked by the language carrier. |

Reviewer 3 Report
The manuscript investigated the compositionally disordered ceramic (Gd,Y,Tb,Ce)3Al2Ga3O12 phosphor for an effective conversion of isotopes' ionizing radition to the light. There are some interesting observations. However, ambiguous points still need to be clarified and revised. The comments are as follows:
1. There are many English grammatical and usage mistakes. I suggest it should be revised by a native English speaker thoroughly.
2. In Materials and Methods: I suggest the preparation of powders and ceramics (milling, forming, and sintering) should be described briefly or added in the supplementary materials.
3. Line 154: I suggest the band contributed from Ce or Tb should be marked in XRL pattern.
4. The explanation and discussion of Figure 6 should be complemented. The physical meanings in the differences of TSL should be explained in detail, and why chose samples 4 and 5?
5. The explanation and discussion of Figure 7 should be complemented. Why there are significant differences between (a) and (b)? I suggest the author should explain why the sample with Ce=0.05 exhibited a different behavior.
6. Why was Table 2 added to the manuscript? The relationship between Table 2 and this study should be explained and discussed.
There are many English grammatical and usage mistakes. I suggest it should be revised by a native English speaker thoroughly.
Author Response
|
The Reviewer comment |
Author response |
|
There are many English grammatical and usage mistakes. I suggest it should be revised by a native English speaker thoroughly.
|
The manuscript has been checked by the language carrier. |
|
In Materials and Methods: I suggest the preparation of powders and ceramics (milling, forming, and sintering) should be described briefly or added in the supplementary materials.
|
New sentences, which are devoted to the preparation of powders, have been added in the revised manuscript.
|
|
Line 154: I suggest the band contributed from Ce or Tb should be marked in XRL pattern |
A new sentence explains that the spectrum is a superposition of Ce3+ and Tb3+ ion bands: a wide Ce3+ interconfiguration luminescence overlaps narrow intraconfiguration luminescence bands of Tb3+ ions.
|
|
The explanation and discussion of Figure 6 should be complemented. The physical meanings in the differences of TSL should be explained in detail, and why chose samples 4 and 5?
|
It is Figure 7 in the revised manuscript. The most important information from the figure is that the TSL intensity was reduced after irradiation. Electrons of selected energy may damage the lattice due to the displacement of atoms. Perhaps some of these knocked-out atoms diffuse and fill lattice defects created during the samples preparation. This observation should be studied carefully and will be the subject of a separate publication.
|
|
The explanation and discussion of Figure 7 should be complemented. Why there are significant differences between (a) and (b)? I suggest the author should explain why the sample with Ce=0.05 exhibited a different behavior.
|
It is Figure 8 in the revised manuscript. A new sentence with the requested comment is included. |
|
Why was Table 2 added to the manuscript? The relationship between Table 2 and this study should be explained and discussed.
|
This table demonstrates that activities and energy deposits from the most popular isotopes suitable for alpha- and beta-voltaics are comparable with the irradiation conditions used in the article. An appropriate comment is included in the article.
|

Round 2
Reviewer 1 Report
The document is ready for publication in the present form.
Reviewer 3 Report
The manuscript has been revised to meet the partial requirements of the comments. However, I suggest the manuscript still needs extensive English revision.
I suggest the manuscript still needs extensive English revision.